# Improving Drug Sensitivity of HIV-1 Protease Inhibitors by Restriction of Cellular Efflux System in a Fission Yeast Model

**DOI:** 10.3390/pathogens11070804

**Published:** 2022-07-16

**Authors:** Jiantao Zhang, Qi Li, Shigehiro A. Kawashima, Mohamed Nasr, Fengtian Xue, Richard Y. Zhao

**Affiliations:** 1Department of Pathology, University of Maryland School of Medicine, Baltimore, MD 21201, USA; jiantao.zhang@som.umaryland.edu (J.Z.); qi.li@som.umaryland.edu (Q.L.); 2Graduate School of Pharmaceutical Sciences, The University of Tokyo, Bunkyo-ku, Tokyo 113-0033, Japan; skawashima@mol.f.u-tokyo.ac.jp; 3Drug Development and Clinical Sciences Branch, Division of AIDS, NIAID, National Institutes of Health, Bethesda, MD 20892, USA; mnasr@niaid.nih.gov; 4Department of Pharmaceutical Sciences, University of Maryland School of Pharmacy, Baltimore, MD 21201, USA; fxue@rx.umaryland.edu; 5Department of Microbiology-Immunology, University of Maryland School of Medicine, Baltimore, MD 21201, USA; 6Institute of Human Virology, University of Maryland School of Medicine, Baltimore, MD 21201, USA; 7Institute of Global Health, University of Maryland School of Medicine, Baltimore, MD 21201, USA; 8Research & Development Service, VA Maryland Health Care System, Baltimore, MD 21201-1192, USA

**Keywords:** fission yeast (*Schizosaccharomyces pombe*), small molecule drug uptake, spheroplasts, electroporation, drug efflux pumps, ABC and MFS transporters, human immunodeficiency virus type 1 (HIV-1), protease (PR), protease inhibitor (PI) drugs

## Abstract

Fission yeast can be used as a cell-based system for high-throughput drug screening. However, higher drug concentrations are often needed to achieve the same effect as in mammalian cells. Our goal here was to improve drug sensitivity so reduced drugs could be used. Three different methods affecting drug uptakes were tested using an FDA-approved HIV-1 protease inhibitor (PI) drug Darunavir (DRV). First, we tested whether spheroplasts without cell walls increase the drug sensitivity. Second, we examined whether electroporation could be used. Although small improvements were observed, neither of these two methods showed significant increase in the EC_50_ values of DRV compared with the traditional method. In contrast, when DRV was tested in a mutant strain PR836 that lacks key proteins regulating cellular efflux, a significant increase in the EC_50_ was observed. A comparison of nine FDA-approved HIV-1 PI drugs between the wild-type RE294 strain and the mutant PR836 strain showed marked enhancement of the drug sensitivities ranging from an increase of 0.56 log to 2.48 logs. Therefore, restricting cellular efflux through the adaption of the described fission yeast mutant strain enhances the drug sensitivity, reduces the amount of drug used, and increases the chance of success in future drug discovery.

## 1. Introduction

Fission yeast (*Schizosaccharomyces pombe*) has been used as a model system to study various fundamental biological processes and human diseases, such as cell cycle regulation and cancers [1,2,3]. It has also been used as a cell-based surrogate system for genome-wide searches of therapeutic antiviral targets [4,5,6,7] and as a cell-based platform for high-throughput screening (HTS) of antiviral drugs and testing [3,8,9,10]. However, a logistic challenge in the discovery and testing of small molecule drugs (SMDs) is that it normally requires a higher drug concentration in fission yeast to achieve the same effect as it does in mammalian cells [11]. The objective of this study was to find ways to improve the drug sensitivity of SMDs in fission yeast.

When an SMD is added to a yeast or mammalian cell culture, it typically crosses the cell membrane by passive diffusion [12]. The SMD diffuses into cells following its concentration gradient without the requirement of energy [12]. One of the differences between fission yeast and mammalian cells in drug uptake is that fission yeast has a thick cell wall. The fission yeast cell wall is a rigid exoskeletal structure that is outside of the plasma membrane [13], whereas a mammalian cell only has a plasma membrane. An old premise is that the fission yeast cell wall could be a physical barrier to SMD when it crosses the cell membrane via passive diffusion, especially when the SMD has a large molecular weight. The fission yeast cell wall is composed of interlinked polysaccharides, such as β-(1,3)-glucan, and galactomannoproteins that form the overall fabric of the cell wall [14]. The cell wall keeps the shape of the yeast cell, protects the cell from mechanical injuries, and prevents it from bursting due to internal turgor pressure [14]. However, fission yeast cell walls can be removed by enzymatic digestion without significantly jeopardizing cell viability, at least for a short period of time [15,16]. The cell wall is restored as soon as the cells start to grow and divide. When cells lose cell walls, the cells will round up to form spheroplasts. At this stage, a higher efficiency of plasmid DNA transformation can be achieved than using cells with cell walls [15,17]. Moreover, spheroplasts could remain viable for months if they are maintained in a hyperosmotic medium to equilibrate the internal turgor pressure and avoid cell lysis [14,15,18,19]. For example, 1.2 M sorbitol, not glycerol or DMSO, could be used to store spheroplasts for up to three months without significant loss of its DNA transformation efficiency [15]. This opens the possibility that SMD uptake could potentially be improved by removing the physical barrier of the cell wall. This possibility is being tested in this study.

Electroporation is another method that could enhance cellular uptake of SMDs through the so-called facilitated diffusion [20]. It works by applying cells with short and high-voltage pulses to overcome the barrier of the cell membrane, thus allowing the uptake of SMDs more easily. Electroporation is commonly used to increase DNA transfection efficiency of plasmid DNA or small interference RNA (siRNA) in tissues and cell cultures [21]. It is also a common method used in fission yeast for a high efficiency of plasmid DNA transformation [22,23]. Conceivably, electroporation could also be used to facilitate the uptake of SMDs and to increase drug sensitivity in fission yeast.

Besides passive diffusion, SMDs could also cross the cell membrane by active transport through cellular proteins that regulate cell membrane activities, such as ion channels or cellular influx [12]. In contrast to passive diffusion, however, active transport requires energy to regulate the influx and efflux of SMDs. Fission yeast has a total of 11 ATP-binding cassette (ABC) proteins that transport a wide variety of substrates, including sugars, ions, lipids, and proteins, across membranes. All these ABC transporters are membrane-associated protein pumps, and most significantly, many of them are also dispensable for cell viability [24]. The dispensability of those ABC transporter proteins creates an opportunity to generate genetic mutants to block the efflux pumps and, thus, to increase the availability of SMDs and, thus, increase the drug potency [24]. Indeed, Bfr1 and Pmd1 are two key ABC efflux pumps [24,25]. Deletions of the *bfr1* and *pmd1* (*bfr1*∆ and *pmd1*∆) genes result in the increase in drug sensitivity of several commonly used cellular inhibitors, such as Brefeldin A (BFA), a drug that inhibits protein transport from the endoplasmic reticulum (ER) to the Golgi complex [25]. Similarly, among 49 possible major facilitator superfamily (MFS) transporters, Mfs3 and Caf5 serve as SMD influx and efflux pumps. When they were both deleted, it also increased the drug sensitivity [26]. Systematic studies of the ABC transporters and MFS transporters suggested that two ABC transporters (Bfr1 and Pmd1), two MFS pumps (Mfs1 and Caf5), and a transcription factor (Pap1) are the major contributors that regulate drug efflux in fission yeast [26]. Subsequently, two enzymes—Erg5, a C-22 sterol desaturase, and Dnf2, a P4-ATPase—were found to cooperatively work with this five-gene complex [26] and regulate the drug efflux [27]. As a result, an MDR-supML (multi-drug-resistance-suppressed and marker-less, also known as SAK836) fission yeast strain that contains the seven-gene deletions was created, which is sensitive to a wide range of bioactive SMDs, including nocodazole and Velcade [27].

In our previous studies, we developed a fission yeast model system to study wild-type and multi-drug-resistant HIV-1 proteases (PRs) [28,29]. In addition, we also developed a fission yeast cell-based HTS system for drug screening of HIV-1 protease inhibitors (PIs) [9]. While HIV-1 PRs cleave the natural HIV-1 substrates and induce apoptotic cell death in the same dose-dependent manner in fission yeast as they do in human cells [28,29], one of the technical limitations was that, very often, we had to use higher drug concentrations to achieve the same effect as in mammalian cells [11]. To improve the drug sensitivity of HIV-1 PI drugs in fission yeast, in this study, we compared the EC_50_ of an FDA-approved HIV-1 PI drug Darunavir (DRV, Prezista™) measured by the traditional passive diffusion method with three different methods as described above, i.e., electroporation, spheroplasts, and the inhibition of efflux pumps by the adapted use of an efflux-pump-defective SAK836 strain [27]. A PR836 fission yeast strain, a derivative of SAK836, that carries a single and integrated copy of the HIV-1 *PR* gene in the yeast chromosome was generated for this study. Like the efflux-pump-defective SAK836 strain, the PR836 strain also carries the same seven-gene deletions as SAK836, including two ABC (*bfr1* and *pmd1*) and two MFS *(mfs1* and *caf5*) genes, along with a transcription factor *pap1* and two enzymes (*erg5* and *dnf2*) that regulate drug efflux [27]. All the current FDA-approved and single-formulation PI drugs were also tested in the PR836 strain and their EC_50_ values were compared with that generated from the wild-type RE294 strain.

## 2. Results

### 2.1. Determination of EC_50_ of DRV against HIV-1 PR by Adding Drug Directly to Cell Culture through Passive Diffusion

A wild-type fission yeast RE294 strain (Table 1) that carries a single and integrated copy of the HIV-1 *PR* gene in the yeast chromosome [9,28] was used in this experiment as a control. An FDA-approved HIV-1 PI drug DRV was chosen because it is one of the most potent inhibitor drugs among its class, both in mammalian cells [30] and in the fission yeast cell model system [11,28].

Since the HIV-1 *PR* gene is under the control of a fission yeast *nmt1* (no message in thiamine) promoter [31,32], the HIV-1 PR protein-specific effect can be measured only after the inducible production of the PR protein. We previously showed that the expression of HIV-1 *PR* prevents the ability of fission yeast cells to form colonies on agar plate and blocks cell proliferation in liquid culture [28]. Here, we determined the inhibitory effect of DRV on the HIV-1 PR effects in RE294 cells.

To establish a baseline of the DRV drug sensitivity as measured by half-maximal effective concentration (EC_50_) for the proposed studies, here, we tested again the inhibitory effect of DRV on HIV-1 PR-induced growth arrest. Growth of the RE294 cells was compared over a period of 72 h post-gene induction (*pgi*) between PR-producing (PR-on) and PR-repressing (PR-off) conditions. As shown in Figure 1A-a, the RE294 cells without HIV-1 PR protein production grew normally with typical exponential growth kinetics. Conversely, no growth was observed in the PR-producing cells over the same time. However, adding 3.0 μM of DRV in the same *PR* gene-inducing culture completely reversed HIV-1 PR-induced growth arrest. To measure the drug sensitivity of DRV, the EC_50_ of DRV was determined in the RE294 strain under the same experimental conditions as Figure 1A-a, but a total of nine different DRV concentrations (0, 0.1, 0.3, 1.0, 3.0, 10, 30, 100, and 150 μM) were added to the PR-expressing cells, and their growth was measured at 72 h *pgi*. The calculated EC_50_ was 2.09 ± 0.18 μM.

**Table 1 pathogens-11-00804-t001:** Fission yeast strains and plasmids used in this study.

Strains or Plasmids	Genotype and Characters	Source or Reference
Fission yeast strains		
SP223	Wild-type; *h^-^, ade6-216, leu1-32, ura4-294*	Lab collection
RE294	*h^-^, ade6-216, leu1-32, ura4-294*::*nmt*1-*PR*(NL4-3)-kanMX; a derivative of SP223 carrying a single integrated copy of HIV-1 *PR* gene under a nmt1 promoter at the ura4 locus	[9,28]
SAK1	Wild-type, *h*^90^, *ade6-M216*, *leu1-32*, *ura4-D18*; a derivative of JY878	[27]
SAK836	Also known as MDR-supML; *h*^90^, *ade6-M216*, *leu1-32*, *ura4-D18*, *caf5*::bsd, ∆*pap1*, ∆*pmd1*, ∆*mfs1*, ∆*bfr1*, ∆*dnf2*, *erg5*::*ura4*^+^	[27]
PR836	*h*^90^, *ade6-M216*, *leu1-32*, *ura4-D18*::*nmt*1-*PR* (NL4-3)-ble1, *caf5*::bsd, ∆*pap1*, ∆*pmd1*, ∆*mfs1*, ∆*bfr1*, ∆*dnf2*, *erg5*::*ura4*^+^; a derivative of SAK836 carrying a single integrated copy of HIV-1 *PR* gene under a *nmt1* promoter at the *ura4* locus	This study
Plasmids		
pYZ1N	A fission yeast expression vector with an inducible *nmt1* promoter and a *leu1* selectable marker	[32]
pYZ1N-PR	Wild-type HIV-1 *PR* gene cloned in pYZ1N	[28]
pJZBle1-PR	A modified pClone1Ble1 vector carrying *ura4* flanking sequence and wild-type HIV-1 *PR* gene driven by a *nmt1* promoter. This plasmid was used to create the PR836 strain.	This study

Note: Bsd, Blasticidin S deaminase; Pap1, a transcription factor; 2 ABC transporters: Bfr1, plasma membrane brefeldin A efflux transporter, and Pmd1, a leptomycin transmembrane transporter Pmd1; 2 MFS transporter: Caf5, plasma membrane spermine family transmembrane transporter Caf5, and Mfs1, plasma membrane xenobiotic transmembrane transporter Mfs1; Dnf2, plasma membrane phospholipid-translocating ATPase complex, ATPase subunit Dnf2; Erg5, C-22 sterol desaturase Erg5.

### 2.2. Effect of Electroporation on the Drug Sensitivity of DRV

The general principle of electroporation is to expose cells to a short and high-voltage pulse that presumably overcomes the barrier of the cell membrane by creating a transient and permeabilized state for DNA or RNA uptake [20]. The same electroporation method can also be used to transfer peptides and small molecules into mammalian cells and yeast cells [20,21,33]. Since electroporation has been shown to improve DNA transformation efficiency in fission yeast cells [22,23], here, we tested whether electroporation can improve the uptake of DRV and further improve its drug potency.

About 1.0 × 10^7^ active growing RE294 cells were prepared the same way as described in Section 2.1. However, the cells were collected here by centrifugation and washed three times with ice-cold sterilized 1.2 M sorbitol instead of dH_2_O. DRV was added to the resuspended cells at the indicated concentration. Ten minutes after incubation on ice, electroporation was conducted on the ECM-600 electroporator (BTX, Holliston, Massachusetts) following the manufacturer’s protocol. The electroporated cells were then diluted to 4 × 10^4^ cells/mL with Pombe Glutamate (PMG) medium containing DRV at the same concentration and incubated at 30 °C for 72 h. The same experimental procedures as described in Section 2.1 were also used to compare the effect of DRV on HIV-1 PR-induced growth arrest in the electroporated cells with that of controls (Ctrl), i.e., DRV was added directly to the growth medium. As shown in Figure 1A-a,A-b, very similar growth kinetics were seen between PR-off and PR-on cells as in the controls. As expected, DRV delivered through electroporation in the final concentration of 150 μM also fully suppressed PR-induced growth arrest. However, a side-by-side comparison of the DRV effect at various drug concentrations (1, 3, 10, and 30 μM) between the electroporated cells and the control cells showed a slight improvement in the suppressive DRV effect. The difference was clearly shown in cells treated with 3 μM and 10 μM of DRV. In the electroporated cells, 3 μM of DRV was sufficient to reach maximum suppression, whereas it took 10 μM in the controls to reach the same level of suppression (Figure 1A-c). Indeed, the calculation of the EC_50_ between the electroporated cells (1.56 ± 0.03 μM) and the control cells (2.24 ± 0.18 μM) showed a 30.36% improvement in the DRV effect in electroporated cells (Figure 1A-d). However, statistical analysis showed no statistical significance between the two groups (*p* > 0.05, paired *t*-test; Graphpad Prism 9.3). Therefore, the results of these experiments showed that delivery of DRV by electroporation to the RE294 cells showed a slight increase but no significant improvement of the DRV drug potency in comparison with that by simple diffusion.

### 2.3. Determination of the Drug Sensitivity of DRV in Spheroplasts

Fission yeast has a thick and rigid cell wall [13] that could potentially be a physical barrier to the delivery or uptake of SMDs. The goal of this experiment was to test whether the delivery of DRV can be improved with spheroplasts by removing the fission yeast cell wall. The method we used to generate spheroplasts has been previously described [16]. Briefly, RE294 cells were grown to 6–8 × 10^6^ cells/mL in 10 mL EMM medium. The cells were harvested and washed twice with 10 mL SCS buffer (20 mm sodium citrate, 1 M d-sorbitol, pH 5.8). Then, the cells were resuspended in 0.5 mL SCS buffer containing 0.1 g/mL of Lallzyme MMX (Lallemand, Montreal, Canada) and incubated at room temperature for 15–20 min. When greater than 95% of cells rounded up, an indication of forming spheroplasts, the resulting spheroplasts were centrifuged (170× *g*, 3 min) and gently washed once with the EMM medium containing 1 M sorbitol. Since spheroplasts are osmotically sensitive [18,34], to avoid bursting of the spheroplasts, we tested various concentrations of sorbitol, a reagent that maintains osmotic pressure, to see which level of sorbitol is the best in maintaining spheroplasts. Ten-fold dilutions of sorbitol were tested with various numbers of spheroplasts. We found that, at 1.0 × 10^5^ cells/mL of spheroplasts, 0.01 M of sorbitol resulted in bursting of about 20% of spheroplasts. In contrast, sorbitol concentrations higher than 2.66 M caused the shrinking of the spheroplasts due to hypotonic intracellular pressure. Sorbitol in the range of 0.1 M to 1.0 M showed neither bursting nor shrinking of the spheroplasts (data not shown).

Thus, we decided to compare the effect of 0.1 M and 1.0 M sorbitol on the drug sensitivity of DRV. DRV with indicated concentration was added to freshly generated spheroplasts (1 × 10^5^ cells/mL) and cultured in 0.1 M or 1.0 M sorbitol-containing EMM media and incubated overnight at 25 °C to allow re-formation of the fission yeast cell walls. The DRV-containing cell cultures were then further grown at 30 °C for 72 h. In the same way as we did in the electroporation tests, the suppressive effect of DRV on HIV-1 PR was determined by comparing the PR-off and PR-on cells with DRV added directly to the spheroplasts and measured by cellular growth over time (Figure 1B-a), by measuring the dose-dependent drug effect at 72 h *pgi* (Figure 1B-c), and by the determination of the EC_50_ (Figure 1B-d). As a result, spheroplasts cultured in 0.1 M sorbitol without the PR production showed similar growth kinetics as the control method (Figure 1A-a, left). However, a significant delay was seen in the growth of spheroplasts cultured in 1.0 M sorbitol (Figure 1B-a, right). Normally, it takes about 24 h *pgi* to show the different growth effects between PR-off and PR-on cultures, as the PR protein is fully produced at about 16 h *pgi* [31,32]. These growth kinetics were indeed seen in the 0.1 M sorbitol culture. However, there was an almost 24 h delay of cellular growth in the 1.0 M sorbitol culture. The growth difference between the PR-off and the PR-on culture diverged at about 48 h *pgi* (Figure 1B-a, right). In addition, spheroplasts in 0.1 M sorbitol also showed slightly stronger suppression of DRV on HIV-1 PR than that in the 1.0 sorbitol medium. For example, when 3.0 μM of DRV was added, the maximum suppression of DRV on the PR at 64 h was at about 59.4 ± 21.1% in the 1.0 M sorbitol culture, whereas close to 93.3 ± 5.3 percentage suppression was seen in the 0.1 M sorbitol culture (Figure 1B-a,B-b). Consistently, it took about 10 μM DRV to reach the maximum suppression in the 1.0 M sorbitol compared with 3 μM of DRV in the 0.1 M sorbitol medium (Figure 1B-c). A comparison of the EC_50_ between the 0.1 M sorbitol cells (1.94 ± 0.09 μM) and the 1.0 M sorbitol cells (2.37 ± 0.13 μM) showed an 18.1% increase in the DRV effect (Figure 1B-d). Therefore, spheroplasts cultured in 0.1 M sorbitol showed a slight improvement in the drug sensitivity of DRV than that in 1.0 M. However, spheroplasts overall did not significantly improve the drug potency of DRV.

### 2.4. Enhanced Drug Sensitivity of FDA-Approved PI Drugs in a Fission Yeast Mutant Strain That Is Defective in Efflux Pumps

A fission yeast mutant SAK836 strain that is defective in cellular efflux pumps was shown to significantly enhance drug sensitivity of a wide range of bioactive SMDs [26,27]. Here, we wanted to test whether the drug sensitivity of DRV could also be improved in this mutant strain. To ensure that the two wild-type and parental fission yeast strains were isogenic, we first examined whether the drug sensitivity of DRV was comparable in both the wild-type fission yeast SP223 strain and the wild-type SAK1 strain. The SAK836 mutant strain was generated from the SAK1 strain [26,27]. As shown in Figure 1C-a, the two EC_50_ curves were nearly indistinguishable, suggesting these two yeast strains were indeed isogenic, at least in their responses to DRV. Interestingly, we noticed that the EC_50_ values of DRV shown in both wild-type strains were about one log higher than what we saw in RE294 (Figure 1C-b). Since the HIV-1 *PR* gene in RE294 is stably integrated into the chromosome as a single copy, we tested whether the observed difference in the EC_50_ was due to the difference of episomal *PR* gene expression from integrated *PR* gene expression. We measured the EC_50_ values in the same SP223 genetic background. Indeed, when HIV-1 PR was produced from the plasmid pYZ1N-PR, the EC_50_ was 25.64 ± 1.59 μM, while the EC_50_ remained at 2.09 ± 0.18 μM. Thus, the RE294 strain was about one log more sensitive than when HIV-1 PR was produced from a plasmid (Figure 1C-b).

As an integrated *PR* gene expression significantly improved the drug sensitivity of DRV, next, we tested whether a single and integrated copy of HIV-1 *PR* in the SAK836 mutant background could further enhance its sensitivity to DRV. A new fission yeast strain (PR836) was created in the same SAK836 genetic background, but it carried a single integrated copy of the HIV-1 *PR* gene under the *nmt1* promoter in the chromosome (Appendix A). Therefore, the PR836 strain, like the SAK836 mutant strain, was also defective in drug efflux pumps [26]. We tested the drug sensitivity of DRV in three different ways, i.e., the HIV-1 *PR* gene was expressed (i) through a pYZ1N-PR plasmid in the wild-type SAK1 strain, (ii) through a pYZ1N-PR plasmid in the mutant SAK836 strain, and (iii) through a single and integrated copy of the HIV-1 *PR* gene from the chromosome of the SAK836 strain. As shown in Figure 1C-c, the episomal expression of the *PR* gene in the SAK836 strain increased the drug sensitivity of DRV by 13.24-fold (1.12 log). An additional 4.43-fold (0.65 log) increase was seen when the *PR* gene was expressed as a single and stably integrated gene from the chromosome.

We previously described the drug sensitivities of FDA-approved PI drugs by measuring their EC_50_ in the wild-type RE294 strain [11]. For the purpose of comparison, here, we measured EC_50_ values of the same PI drugs in the mutant PR836 stain, and the results are compared side-by-side in Table 2. The differences in the EC_50_ values generated from the same drug between the RE294 and the PR836 strains are listed in fold changes and in log scales. The statistical significance of the difference between each pair was calculated by the pair-wise *t*-test analysis. The chemical structures of those PI drugs are shown in Appendix A. Consistent with our previous report [11], the EC_50_ values of these PI drugs in the RE294 strain showed a wide range of about 1.74 logs in drug sensitivities, with DRV as the most potent drug (EC_50_ = 2.09 ± 0.18 μM) and RTV as the least strong drug (EC_50_ = 115.2 ± 3.20 μM). In the PR836 strain, however, the difference in EC_50_ values among these drugs were much narrower (0.88 logs) than in the RE294 strain, with one outlier of Indinavir (IDV). The most potent PI in the PR836 strain was Lopinavir (LPV; EC_50_ = 0.10 ± 0.01 μM), with IDV as the least potent drug (EC_50_ = 1.52 ± 0.16 μM). When comparing the EC_50_ values of the same drug between the RE294 and the PR836, there were clear and statistically significant differences between the two cell systems based on the pair-wise *t*-test analysis. Overall, the EC_50_ observed in the PR836 showed a marked improvement in EC_50_. 

For example, the smallest increase was 3.6-fold (0.56 log; *p* < 0.05), which was seen in DRV, whereas a more than 300-fold (2.48-log; *p* < 0.01) increase in the EC_50_ was observed in RTV.

In summary, when tested in the PR836 strain, all the FDA-approved and single-formulation PI drugs tested here showed significantly higher drug sensitivities, ranging from 0.56 log to 2.48 logs, than that in the RE294 strain.

## 3. Discussion

The goal of this study was to find ways to increase the drug sensitivity of the FDA-approved HIV-1 PI drugs in the fission yeast model system that we previously developed [9,28]. The drug sensitivity was determined by measuring the EC_50_ and tested by using electroporation, spheroplasts, and a fission yeast mutant strain that is defective in drug efflux [26]. Among the three different methods tested, electroporation showed a small but not statistically significant increase (30.3%) in the EC_50_ of DRV in comparison with the control simply adding the drug directly to the cell culture through passive diffusion (Figure 1A). Although spheroplasts cultured in 0.1 M sorbitol showed a stronger (18.1%) suppressive effect of DRV on the HIV-1 PR over spheroplasts in 1.0 M, overall, spheroplasts did not significantly improve the drug potency of DRV (Figure 1A-b). In contrast, significant improvement in the DRV potency was observed when the inhibitory effect of DRV on HIV-1 PR was tested in a fission yeast mutant PR836 strain, in which some of the key proteins that regulate drug efflux pumps were deleted [26,27] (Figure 1A-c). Testing the remaining eight FDA-approved HIV-1 PI drugs showed an even higher increase in drug sensitivities, which overall ranged from 0.56 log to 2.48 logs (Table 2).

The strategy we used to improve the drug sensitivity of the HIV-1 PI drugs was to increase the drug uptake of the target cells and the intracellular drug concentration that is required to show the inhibitory effect of the PI drug on HIV-1 PR. As described in the introduction, drug uptake of an SMD could be accomplished by passive diffusion or by active transport with the aid of methods such as electroporation, increased drug influx, or limiting drug efflux [12].

As electroporation has been successfully used to improve plasmid DNA transformation efficiency by increasing the uptake of plasmid DNA [22,23], we suspected that a similar improvement in SMD uptake might also be observed. However, the electroporation only generated marginal enhancement of the DRV effect in comparison with the traditional passive diffusion method (Figure 1A-a). Nevertheless, it is still possible that electroporation may have increased the uptake of DRV. However, unlike the plasmid transformation where the plasmid can be selected by antibiotic or auxotrophic selection markers post-electroporation, no selection was available to retain the DRV within cells. Instead, after the DRV entered cells, some of it could have been pumped out again by an active drug efflux transporting system.

The fission yeast cell wall was thought to be a physical barrier to the uptake of SMDs. Contrary to this belief, however, our data showed that the uptake of DRV did not significantly improve in the fission yeast cells when cell walls were removed by enzymatic digestion (Figure 1A-b). Like the electroporation method, plasmid DNA transformation efficiency can be significantly improved in the wall-less spheroplasts [15,17]. Thus, we surmised that the uptake of DRV might have been improved in spheroplasts. However, some of the DRV taken up may have been lost due to the lack of a selection marker and, thus, been pumped out by the effective drug efflux system. Nevertheless, our data confirmed that the use of sorbitol in the range of 0.1 M to 1.0 M could be used to keep the spheroplasts viable without bursting or shrinking them. Furthermore, we showed that spheroplasts in 0.1 M sorbitol are slightly better than 1.0 M of sorbitol in showing the suppressive effect of DRV on HIV-1 PR. However, spheroplasts overall did not significantly improve the drug potency of DRV. Therefore, fission yeast cell walls may not be the major barrier to DRV uptake, but rather, the balance between drug influx and efflux may play an important role in maintaining the intracellular drug concentration. Indeed, significant improvement in the DRV drug potency was observed in a drug-efflux-defective PR836 strain (Figure 1C).

The PR836 strain is a derivative of the SAK836 strain, which is defective in drug efflux pumps [26]. To create such a drug-efflux-defective strain without jeopardizing the cell viability, a total of seven genes were deleted, including two ABC transporters (*brf1* and *pmd1*), two MFS transporters (*mfs1* and *caf5*), a transcriptional factor *pap1* that supports drug efflux, and two enzymes (*efg5* and *dnf2*) that are required to maximize SMD sensitivity [26,27]. In the original study, a wide range of bioactive SMDs showed significant enhancement of drug sensitivity in the SAK836 strain [26,27]. For example, they screened a LOPAC 1280 SMD library in the wild-type SAK1 and the SAK836 mutant strain and measured how many SMDs inhibit >80% of cellular growth at 20 μM in each strain. A total of 51 SMDs were detected in the wild-type strain, whereas 132 SMDs showed the same level of inhibition, indicating an overall increase in drug sensitivity, in the SAK836 strain [26,27].

In this study, the PR836 strain was created in the same mutant genetic background as SAK836, but it also carries a single and integrated copy of the HIV-1 *PR* gene from the chromosome (Table 1; Appendix A). Consistent with the idea that limiting the drug efflux system in the SAK836 increases the drug sensitivity, we indeed saw a marked improvement in the EC_50_ in all the HIV-1 PI drugs tested when compared with that in the RE294 (Table 2). The enhancement of PI drug potency ranged from an increase of 0.56 log in DRV to 2.48 logs in RTV. It is unclear, however, why there is such a wild range of drug sensitivity of a PI drug between the wild-type and the mutant strain, especially because these PI drugs have similar molecular weights and hydrophobicity. One possible explanation is that RTV, as an example, was designed to mimic the natural HIV-1 PR substrates by competitive fitting to the active enzymatic site [35]. Since RTV is competing with its natural substrates, there may be free RTV within cells to be pumped out in the wild-type yeast strain. Conversely, when the effective drug export system is significantly restricted in the PR836 mutant, we would expect more intracellular RTV to be retained within cells and, thus, show a much greater improvement in EC_50_ value. In contrast, even though DRV is also a competitive inhibitor, it is a P2-ligand drug that binds to the S2/S2’ backbone atoms of HIV-1 PR with the maximum hydrogen-bonding affinity. This high-affinity binding prevents the PR from accessing its natural viral substrates [36]. As a result, there may be little or no DRV left within cells to be exported by the drug efflux pumps, regardless of whether it is in the wild-type or the mutant cells [36].

If these explanations were true, the observed discrepancies in drug sensitivities would confirm the idea that, besides the drug uptake, the resulting intracellular concentration that is maintained by the balance of drug influx and efflux regulation is also critical in determining the drug sensitivity. In addition, the drug potency also depends upon the copy number of the gene target. For example, a 17.05-fold increase in the EC_50_ value was seen when the DRV effect was tested in the wild-type RE294 strain in comparison with the SP223-PR strain (Figure 1C-b). The RE294 strain carries a single and integrated copy of the HIV-1 *PR* gene, and the HIV-1 *PR* gene in the SP223-PR strain was expressed episomally on a multicopy plasmid. Interestingly, a smaller (4.43-fold) but significant increase in the EC_50_ was also seen when the DRV effect was tested in the SAK836 mutant background against a single copy (PR836) over the multicopy of *PR* gene expression (SAK836-PR). Although there was no report showing that those gene deletions in the SAK836 strain affect the export of plasmid DNA in the fission yeast cells, it is possible that restricting cellular export in this mutant strain also maintains a higher copy number of plasmid than that in the wild-type strain.

Despite the fact that we observed significant improvement in EC_50_ in all the FDA-approved HIV-1 PI drugs tested in PR836 when compared with that in the RE294 (Table 2), not all drug sensitivity of SMDs can be improved by testing them in the SAK836 strain, as demonstrated in a different study [26]. Understandably, not all SMDs cross cell membranes in the same manner. For example, the drug sensitivity of benomyl-based compounds was similar in both the wild-type and the mutant strains, suggesting that those SMDs are not likely to be pumped out by ABC and MFS transporters [26]. Moreover, in fission yeast, depending upon the molecular weights and chemical structure of the SMDs, the difference in drug sensitivity of the same drug between fission yeast and human cells also varies. The difference could range from a similar level to a concentration several logs higher of an SMD to achieve the same effect as it does in mammalian cells [11,28]. However, the difference in drug sensitivity does not affect the functionality of the drug, as all FDA-approved HIV-1 PI drugs effectively inhibited the specific effects of HIV-1 PRs in the same manner as they do in mammalian cells [11]. The highlight of this study is that we now are able to narrow the gap in the use of drug concentrations required to achieve a similar EC_50_ between fission yeast and mammalian cells. For example, the EC_50_ of RTV reported in mammalian cells ranges from 40 to 100 nM [36,37]. In the PR836 cells, the EC_50_ of RTV was about 300 nM, which is on a similar order of magnitude as the mammalian cells.

Moreover, we were able to rank the potency of the tested FDA-approved HIV-1 protease inhibitor drugs in the wild-type of fission yeast strain RE294 (Table 2), which was in a similar order to that shown in human cells [11]. In addition, we also developed a fission yeast cell-based HIV-1 protease enzymatic assay [28,29]. There was a general correlation between the EC_50_ and the inhibitory enzymatic activities of the PI drugs. However, we were not able quantify the correlation, as the enzymatic assay is a semi-quantitative assay that cannot be used to generate IC_50_ [28,29].

In summary, fission yeast is a fitting surrogate system for HTS and testing of antiviral drugs against HIV-1 PR [9,28,29] and other viral targets [7,8]. The advantages of using a yeast cell-based system are that it shares all the same benefits as a mammalian cell-based drug screening system, i.e., (i) cytotoxic compounds are automatically removed from the screenings due to cell death, (ii) it detects both orthosteric and allosteric inhibitors, and (iii) it is fast and easy to culture, making it particularly suitable for HTS. However, a major obstacle to using a fission yeast cell-based system in HTS is that high drug concentrations are often needed. As we demonstrated in this report that the adaption of the mutant SAK836 strain to restrict the cellular efflux transport systems enhances the drug sensitivity and allows the use of much lower concentrations than before, this improvement thus increases the chance of success in future high-throughput SMD screening and testing.

## 4. Materials and Methods

### 4.1. Fission Yeast Strains, Plasmids, and Growth Media

The fission yeast strains and plasmids that were used in this study are summarized in Table 1. Standard YES (Yeast Extract with Supplements), complete or minimal PMG (Pombe Glutamate Medium), or EMM (Edinburgh minimal medium) selective media supplemented with 225 μg/mL of adenine, uracil, leucine, or thiamine (20 µM) was used to grow fission yeast cells or to select for plasmid-carrying cells. Luria–Bertani (LB) medium supplemented with Ampicillin (100 µg/mL) was used for growing *E. coli* DH5α cells and for DNA transformation.

### 4.2. Generation of the Mutant PR836 Strain

The PR836 is a genetically engineered mutant fission yeast strain that is defective in drug efflux pumps [26] and carries a single integrated copy of the HIV-1 *PR* gene under the control of the *nmt1* promoter in the chromosome. The PR836 strain was generated from a multi-drug-resistance-suppressed and marker-less (MDR-supML) fission yeast strain (SAK836) that was created by [26]. The SAK836 was a derivative of the wild-type SAK1 strain, which contains a total of 7 deletions of key proteins that regulate the drug efflux pump, including 2 ABC gene deletions (*brf1* and *pmd1*), 2 MFS gene deletions (*mfs1* and *caf5*), *pap1*, which is a transcriptional factor, and two enzymes (*erg5* and *dnf2*) that are required to maximize the SMD sensitivity [26,27].

To create the PR836 strain carrying a single integrated copy of the *PR* gene at the *ura4* locus of SAK836, we created a new construct named as pJZBle1-PR via the Gibson Assembly method [38]. The upstream (806 bp) and downstream (926 bp) DNA sequences of the *ura4* gene were amplified from the SP223 genome due to the *ura4-D18* genotype of SAK836, in which the 1.8 kb fragment containing whole *ura4* coding sequence and partial *ura4* upstream and downstream sequences had been deleted from the chromosome (Appendix A) [39]. To minimize the insertion size in fission yeast, the fragment containing AmpR expression cassette and origin sequence amplified from pCloneBle1 were placed in between the downstream and upstream DNA sequence of the *ura4* gene. The HIV-1 *PR* expression cassette from pYZ1N-*PR* was inserted via *Hpa* I and *Eco*R I sites. After the sequencing confirmation, the resulting plasmid was linearized by digestion with *Nhe* I and integrated into the SAK836 genome via homologous recombination. The resulting colonies were selected on YES medium plate containing zeocin (300 mg/L; ThermoFisher Cat#: R25001;) and blasticidin (30 mg/L; ThermoFisher Cat#: R21001), which were further verified by PCR using *PR*- or *ura4* flanking sequence-specific primers and yeast growth assays to confirm HIV-1 PR-induced cell growth arrest [11,28]. All the primers used in this study are listed in Appendix A.

### 4.3. Inducible Gene Expression of HIV-1 PR Gene in Fission Yeast

All the experimental methods to induce HIV-1 *PR* gene expression in fission yeast and to measure its activities in fission yeast have been previously described [11,28]. Briefly, all fission yeast cells were grown in a minimal PMG medium supplemented with 225 μg/mL of adenine, uracil, leucine, thiamine, or G418 based on the auxotrophic markers for yeast cell growth and plasmid selections [9,28,29]. RE294 cells were grown in PMG supplemented with adenine, uracil, leucine, and geneticin (G418, 200 mg/L). PR836 cells were grown in PMG supplemented with adenine, leucine, Zeocin (300 mg/L), and blasticidin (30 mg/L). HIV-1 *PR* gene expression in both RE294 and PR836 is under the control of an inducible no message in thiamine (*nmt1)* promoter [31]. Specifically, the starting fission yeast cell culture in the presence of 20 μM thiamine (to prevent *nmt1*-mediated gene expression) was first inoculated and refreshed by overnight growth at 30 °C to obtain active growing cells. Cells were washed three times with distilled water and diluted to a final concentration of 2 × 10^4^ cells/mL in PMG with 20 μM thiamine (PR-off) or without thiamine (PR-on). Full gene expression can be achieved in about 16 h post-gene induction [31,40].

### 4.4. Fission Yeast Growth Assay

The assay used to measure HIV-1 PR-induced cell growth arrest in fission yeast cells has been previously described [28,41]. Briefly, RE294 or PR836 fission yeast cells were grown in the minimal PMG media to the mid-log growth phase in the presence of 20 μM thiamine. Cells were washed three times with distilled water and diluted to a final concentration of 2 × 10^4^ cells/mL in PMG media supplemented with thiamine to suppress HIV-1 PR gene expression or without thiamine induce PR gene expression. Cell culture in the volume of 100 μL/well was dispensed to a 96-well microtiter plate and incubated at 30 °C over time. Cell growth was measured as optical density of 600 nm (OD_600_) at the indicated time using a Synergy H1M microplate reader (BioTeck).

### 4.5. Determination of EC_50_ in Fission Yeast

The method used to determine the EC_50_ of HIV-1 PI drugs in fission yeast has been previously described [11]. Briefly, RE294 or PR836 fission yeast cells were grown in the minimal PMG media as described in Section 4.5. The expression of the HIV-1 *PR* gene was induced by removing thiamine from the growth medium. A PI drug was added to the *PR*-expressing cells in concentrations of 0.01, 0.1, 0.3, 1.0, 3.0, 10, 30, 100, and 150 μM. The final cellular growth of drug-treated cells was measured at 72 h *pgi* by the OD_600_. A dose-dependent growth curve was generated by comparing the result of drug-treated *PR*-producing cells with that of *PR*-repressing control cells to produce the relative % of cellular growth to the PR-off cells. The EC_50_ curve was calculated based on the best-fit curve with a variable slope in GraphPad Prism (Version 9.3).

### 4.6. Electroporation of SMDs in Fission Yeast

Electroporation was performed following the fission yeast plasmid transformation protocol in the ECM-600 operation manual (BTX, Holliston, MA, USA). Briefly, fission yeast RE294 cells were grown in a PMG medium to a cell density of 1 × 10^7^ cells/mL. Cells were collected by centrifugation and washed three times with ice-cold sterilized 1.2 M sorbitol. DRV at the indicated concentration was added into resuspended cells in 1.2 M sorbitol. Ten minutes after incubation on ice, electroporation was conducted on the ECM-600 electroporator with the parameters of: 2.25 kV, 129 Ω, 50 μF. Electroporated cells were then diluted to 4 × 10^4^ cells/mL with PMG medium with DRV at the same concentration and incubated at 30 °C for 72 h.

### 4.7. Transfer of SMDs into Spheroplasts of Fission Yeast

The method to generate fission yeast spheroplasts using Lallzyme has been previously described [16]. Briefly, RE294 cells were grown to 6–8 × 10^6^ cells/mL in 10 mL EMM medium. The cells were harvested and washed twice with 10 mL SCS buffer (20 mm sodium citrate, 1 M d-sorbitol, pH 5.8). Then, the cells were resuspended in 0.5 mL SCS buffer containing Lallzyme MMX (0.1 g/mL; Lallemand, Montreal, Canada) and incubated at room temperature for 15–20 min. When greater than 95% of cells rounded up, an indication of forming spheroplasts, the resulting spheroplasts were centrifuged (170 × *g*, 3 min) and gently washed once with an EMM medium containing 1 M sorbitol. To add a small molecule drug to spheroplasts, freshly generated spheroplasts (1 × 10^5^ cells/mL) were cultured in EMM medium containing 1 M or 0.1 M sorbitol with DRV added at the indicated concentration. Spheroplasts with added DRV were incubated overnight at 25 °C to allow re-formation of the fission yeast cell wall. The DRV-containing cell cultures were then further grown at 30 °C for 72 h.

### 4.8. Small Molecule Drugs

The name and molecular weight of a group of FDA-approved small molecule HIV-1 PI drugs used in this study are summarized in Table 2. All these PI drugs are rationally designed competitive inhibitors against HIV-1 PR [37,42]. The chemical structures of these drugs are shown in Appendix A. The inhibitory effect of each of these PI drugs on HIV-1 PR-mediated enzymatic and cellular activities in fission yeast cells were previously described [11,28,29]. All PI drugs used in this study were provided by the NIH HIV Reagent Program (ARP), Division of AIDS, NIAID, NIH. They were all dissolved in DMSO.

## Figures and Tables

**Figure 1 pathogens-11-00804-f001:**
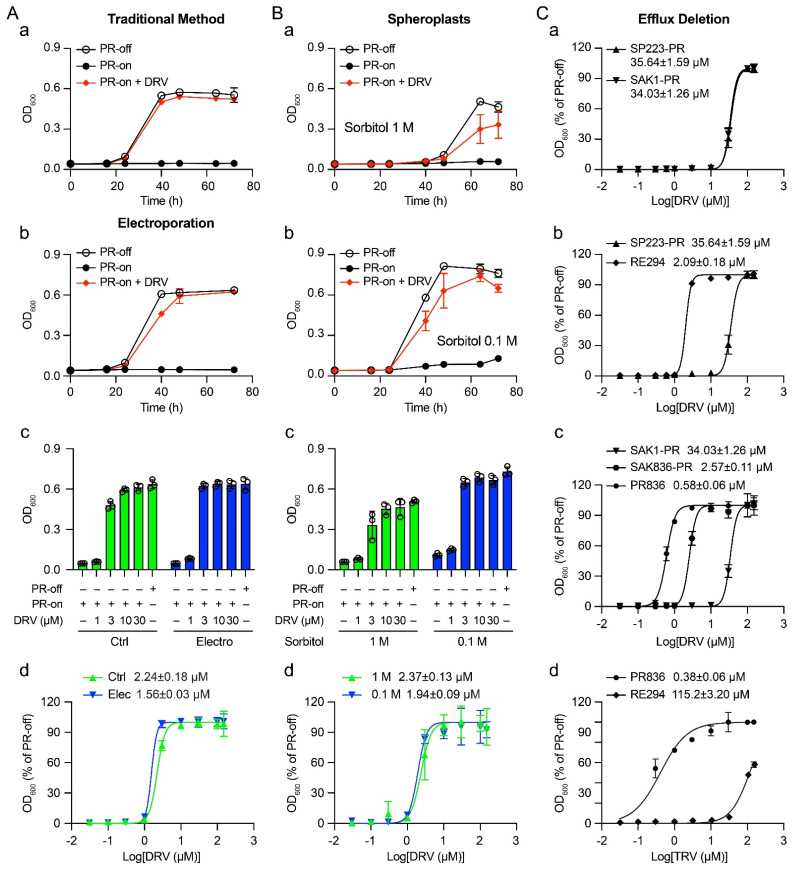
Improving the sensitivity of HIV-1 PR inhibitor drugs in the fission yeast model system using electroporation, spheroplasts, and inhibition of efflux pumps. (**A**) The suppressive effect of HIV-1 PR inhibitor DRV in the fission yeast RE294 strain by delivering the SMD through the traditional method (**a**) and electroporation (**b**–**d**). RE294 strain that carries a single integrated copy of the HIV-1 *PR* gene under a *nmt1* promoter at the *ura4* locus of the chromosome of SP223 was used in these experiments. The suppressive effect of DRV on the HIV-1 PR activities was determined by comparing the PR-off (*PR* gene-repressing), PR-on (*PR* gene-inducing) cells with and without DRV addition. RE294 cells were directly cultured with a growth medium containing 3 μM of DRV (**a**) or electroporated with 3 μM of DRV and maintained in the growth medium with DRV at the same concentration (**b**). An amount of 2 × 10^4^ cells/mL of active growing cell culture was used to compare cellular growth overtime. PR-on + DRV, 3.0 μM of DRV was added to the *PR* gene-inducing culture. (**c**) The dose-dependent drug effect at a fixed time window of 72 h *pgi*. (**d**) The EC_50_ of DRV in the RE294 strain was determined 72 h post-gene induction (*pgi*) by a 9-dose (0, 0.1, 0.3, 1.0, 3.0, 10, 30, 100, and 150 μM) drug treatment scheme. Ctrl, control. (**B**) The suppressive effect of HIV-1 PR inhibitor DRV in spheroplasts of the fission yeast RE294 strain. A total of 9.0 × 10^7^–1.2 × 10^8^ active growing RE294 cells were used for the preparation of spheroplasts as described in the Materials and Methods. The efficiency of cellular delivery of DRV in spheroplasts was first compared between the spheroplasts incubated with 1.0 M and 0.1 M sorbitol. The suppressive effect of DRV on HIV-1 PR was determined by comparing the PR-off and PR-on cells with DRV added directly to the growth medium with the spheroplasts and measured by cellular growth over time (**a**,**b**) with 3 μM of DRV, the dose-dependent drug effect at a fixed time window of 72 h *pgi* (**c**), and the determination of EC_50_ (**d**). (**C**) Enhanced suppression effect of DRV in the fission yeast PR836 strain. The PR836 strain is a new HIV-1 *PR*-carrying strain generated during this study, which is a derivative of a previously established mutant strain (SAK836 or MDR-sup) that is defective in drug efflux pumps [26]. Like RE294, PR836 also carries a single integrated copy of the HIV-1 *PR* gene under a *nmt1* promoter at the *ura4* locus of the chromosome. (**a**) Comparison of the suppressive effect of DRV on HIV-1 PR when it was expressed on a plasmid in SP223 wild-type (SP223-PR) and the parental wild-type of SAK836, SAK1 strain (SAK1-PR). (**b**) Comparison of the suppressive effect of DRV on HIV-1 PR when it was expressed as a single integrated chromosomal copy (RE294) and on a plasmid in SP223 wild-type strain (SP223-PR). (**c**) Comparison of the suppressive effect of DRV on HIV-1 PR when it was expressed as a single integrated chromosomal copy (PR836), on a plasmid (SAK836-PR) in SAK836 mutant strain, and on a plasmid in SAK1 wild-type strain (SAK1-PR). (**d**) Comparison of the suppressive effect of Ritonavir (RTV), the one with the most improved drug sensitivity, on HIV-1 PR between RE294 and PR836. All the EC_50_ curves were calculated based on the best-fit curves with a variable slope in GraphPad Prism (Version 9.3).

**Table 2 pathogens-11-00804-t002:** Comparison of drug sensitivity of FDA-approved HIV-1 PI drugs in the wild-type RE294 and the mutant PR836 fission yeast strains.

Drug Name	Brand Name	Molecular Weight (g/mol) *	EC_50_ in RE294 (μM)	EC_50_ in PR836 (μM)	Difference in EC_50_ (Fold)	Difference in EC_50_ (log10)
Ritonavir (RTV)	Norvir	720.94	115.2 ± 3.20	0.38 ± 0.06	303.2	2.48
Amprenavir (APV)	Agenerase	505.63	55.41 ± 3.18	0.35 ± 0.06	163.0	2.21
Lopinavir (LPV)	Kaletra	628.81	4.59 ± 0.06	0.10 ± 0.01	45.9	1.66
Indinavir (IDV)	Crixivan	613.79	48.64 ± 2.51	1.52 ± 0.16	32.2	1.51
Atazanavir (ATV)	Reyataz	704.86	21.08 ± 0.07	0.75 ± 0.02	28.1	1.45
Saquinavir (SQV)	Invirase	670.84	9.88 ± 0.25	0.74 ± 0.07	13.4	1.13
Nelfinavir (NFV)	Viracept	567.78	9.7 ± 0.34	0.75 ± 0.03	12.9	1.11
Tipranavir (TPV)	Aptivus	602.66	7.86 ± 0.32	0.74 ± 0.07	11.2	1.05
Darunavir (DRV)	Prezista	547.66	2.09 ± 0.18	0.58 ± 0.06	3.6	0.56

Note: The difference in the EC_50_ values is listed in descending order for easy comparison. *, data are from go.drugbank.com.

## Data Availability

Not applicable.

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
