# Peer review of "Improving Drug Sensitivity of HIV-1 Protease Inhibitors by Restriction of Cellular Efflux System in a Fission Yeast Model"

_pathogens, 2022, doi:10.3390/pathogens11070804_

Round 1
Reviewer 1 Report
This study aims to solve the long standing problem in the yeast assay which is the drug efflux. Three methods were explored including deletion of the cell walls, electroporation, and inhibition of drug efflux pump. While neither deletion of the cell walls nor electroporation solved the issue, inhibition of drug efflux by the genetically engineered SAK836 strain. The assay was first calibrated using darunavir, and then expanded to other HIV protease inhibitors. All the tested FDA-approved HIV protease inhibitors showed significantly higher drug sensitivities ranging from 0.56 to 2.48 logs in the PR836 strain than in the RE294 strain. Overall, this is an outstanding study and a significant advance in the field. The PR836 strain is expected to speed up the drug discovery process of HIV protease inhibitors. Comments are:
1. This study mainly focuses on genetic deletion of drug efflux pumps, have the authors thought about using drug efflux inhibitors such as CP-100356?
2. Any explanation why the HIV protease inhibitors showed different EC50 fold changes in the yeast assay? Does this mean some drugs are better drug efflux substrates than others?
3. A growth curve comparison of the RE294 and PR836 without HIV PR expression might be useful to demonstrate that genetic modification did not change the growth kinetics.
4. A comment should be made whether the EC50 values in the yeast assay correlate with their IC50 values in the protease enzymatic assay. A correlation plot might be useful. It is expected that the yeast assay should be able to rank the potency of HIV protease inhibitors. A negative control with a HCV protease inhibitor might further strengthen the results.
